# Prevention of Congenital Cytomegalovirus Infection: Review and Case Series of Valaciclovir versus Hyperimmune Globulin Therapy

**DOI:** 10.3390/v15061376

**Published:** 2023-06-15

**Authors:** Giovanni Nigro, Mario Muselli

**Affiliations:** 1Non-Profit Association Mother-Infant Cytomegalovirus Infection (AMICI), 00198 Rome, Italy; 2Department of Life, Health and Environmental Sciences, University of L’Aquila, 67100 L’Aquila, Italy; mario.muselli@graduate.univaq.it

**Keywords:** congenital CMV infection, prevention hygienic measures, CMV screening, hyperimmune globulin, valaciclovir

## Abstract

Cytomegalovirus (CMV) is the most common cause of congenital infections in developed countries because is capable of infecting the fetus after both primary and recurrent maternal infection, and because the virus may be spread for years through infected children. Moreover, CMV is the most serious congenital infection associated with severe neurological and sensorineural sequelae, which can occur at birth or develop later on. Hygienic measures can prevent CMV transmission, which mainly involve contact with children under 3 years of age and attending a nursery or daycare. In animal and human pregnancies, many observational and controlled studies have shown that CMV-specific hyperimmune globulin (HIG) is safe and can significantly decrease maternal–fetal transmission of CMV infection and, mostly, the occurrence of CMV disease. Recently, valaciclovir at the dosage of 8 g/day was also reported to be capable of decreasing the rates of congenital infection and disease. However, comparing the results of our two recent case series, the infants born to women treated with HIG showed significantly lower rates of CMV DNA positivity in urine (9.7% vs. 75.0%; *p* < 0.0001) and abnormalities after follow-up (0.0% vs. 41.7%; *p* < 0.0001). The implementation of CMV screening would enable primary prevention via hygiene counseling, improve the understanding and awareness of congenital CMV infection, and increase the knowledge of the potential efficacy of preventive or therapeutic HIG or antiviral administration.

## 1. Introduction

Cytomegalovirus (CMV) is the most prevalent cause of congenital infections because CMV is capable of infecting the fetus following both primary and recurrent (reactivation or reinfection) infection in immunocompetent women, and because the virus may be spread for years through infected children [1]. Several “immune evasion” genes allow CMV to escape both humoral and cellular immune responses, contributing to its ability to persist and reactivate in the host. Therefore, primary CMV infection is followed by chronic infection or viral latency, from which the virus can be reactivated (second type of infection). Reinfection by a CMV strain different from the one already infecting a person may also occur, being sometimes symptomatic and associated with an overwhelming detection of CMV DNA. However, both reactivation and reinfection, if they occur via mother to fetus, do not generally cause severe fetal disease. Therefore, a primary infection occurring in a pregnant woman who has never acquired CMV (seronegative) should be distinguished from a woman who already had antibodies before pregnancy (seropositive) [2].

Congenital CMV infection affects 0.2–2.2 percent of all live births, of whom 11–12.7 percent have symptoms including cerebral and sensorineural damage [3,4]. However, the majority of infected neonates acquire CMV from seropositive mothers, or seronegative women who had a primary infection in the second half of pregnancy and have a low occurrence of symptoms at birth and later on [5,6]. In fact, in developing and highly populated countries, there is a high number of seropositive pregnant women, and a higher prevalence of asymptomatic congenital infections than in industrialized countries [7,8]. On the contrary, CMV disease occurs in up to 50% of congenitally infected neonates born to mothers who acquired the infection in the first half of pregnancy [4,5].

Primary CMV infection, which can occur in 1% to 4% of pregnant women, depending on the seroprevalence in each area, is highly dangerous to the fetus when the virus is transmitted in the first four months of pregnancy, when the embryo develops and is especially vulnerable viral insults [2,9]. The transmission rates are remarkably consistent among numerous studies, ranging from 30 to 42%, 38 to 44%, and 59 to 73% for the first, second, and third trimesters, respectively, and increase as pregnancy progresses and the placenta becomes older and more permeable [3,4]. For preconceptional (between 1 and 12 weeks before the last menstrual period) and peri-conceptional (1 week before to 5 weeks after the last menstrual period) infections, the risk of CMV transmission is approximately 6–9% and 19–31%, respectively [10,11]. The reason CMV is transmitted from the children attending a daycare to their mothers is that a low CMV load can infect pregnant women because of the progesterone-linked immunodepression aimed to avoid fetal rejection. Therefore, it is presumable that maternal infections which are considered preconceptional could be immediately post-conceptional [12].

## 2. Pathogenetic Mechanisms

CMV affects all human populations, accounting for around 60% of adults in industrialized nations and more than 90% in many developing countries [1,3]. The adaptation of CMV to the human immune system, and the subsequent two-sided relationship, is critical in understanding viral pathogenicity and the development of clinical manifestations. In immunocompetent hosts, CMV infection results in minor symptomatology, although there is evidence that infected people may have long-term indirect consequences due to the activation of a chronic inflammatory cell-mediated immune response [13]. On the other hand, CMV is the most prevalent and dangerous opportunistic infection following solid organ transplantation or hematopoietic stem cell transplantation, and it continues to be a major opportunistic infection in HIV patients [14,15]. A growing fetus must be considered an immunocompromised host, since half of the genes, and all other organic material, is external to the mother [16]. Therefore, not only transplant recipients or AIDS patients but also fetuses must be considered at risk of a severe or deadly illness, like those undergoing a 50% allograft [12].

Following initial maternal infection, CMV can be transferred to the fetus via viremia and the transmission of infected leukocytes across the placenta, or through an infected placenta (placentitis) and the dissemination to amniotic cells, which are swallowed by the fetus. CMV replication in the fetal oropharynx is followed by hematogenous dissemination to target tissues such as tubular epithelium or salivary cells, which appear to be key sites of replication. Another possible method of transmission is the ascending route from the cervix, which may be infected by CMV [12,17].

Pre-existing humoral immunity protects about 75% of seropositive women against reinfection or reactivation [18]. Neutralizing titers and IgG avidity against CMV are both negatively associated with fetal transmission, although cellular immunity plays an important role in controlling viral transmission and pathogenicity throughout pregnancy [19]. The placenta is a key CMV pathogenic site, which may be affected by its ability to supply oxygen and nutrition to the growing fetus [20]. Primary CMV infection has been linked to placental expansion following viral placentitis in women with fetal or neonatal infection [21]. Although the pathogenic mechanisms of tissue damage are not completely understood, they are as follows: (a) direct tissue injury caused by persistent CMV replication in infected cells, most likely associated with a high viral concentration, resulting in clinically evident disease; (b) ischemic tissue damage (vasculitis) caused by the viral presence in endothelial cells of vessels in several organs, including the placenta and brain; and (c) immunomediated tissue injury caused by immune complex deposition. Meningoencephalitis, periventricular calcifications, microcephaly, polymicrogyria, and other migrational changes in the neurons are the most severe and distinguishing signs of prenatal CMV infection. The closeness of the cerebrospinal fluid (CSF) channel, through which CMV is likely to spread, and the actively reproducing subependymal germinal matrix cells, which are particularly sensitive to CMV, may explain the viral preference for the periventricular region. Microcephaly is associated with encephaloclastic viral effects and potential neuro-proliferative issues caused by CMV interference. Migrational changes indicate that teratogenic consequences of CMV may occur during the first trimester end and the beginning of the second trimester, when neuronal migration occurs. CMV is the only congenitally transmitted infection that causes altered cortical development, with pathogenesis including both teratogenic and encephaloclastic pathways. Hearing loss may result from CMV-inhibited cochlear cell growth or brain damage in early prenatal infections, or from an increasingly chronic infection of the inner ear and auditory nerve [2,18,22].

***Virus–host interaction.*** The body’s relationship with CMV is based on intrinsic, innate, and adaptive immune responses of human beings, as well as numerous viral countermeasures, which collectively enable a dynamic balance between host and pathogen that largely prevents disease while not eliminating the virus from the human body. As with other infections, pathogen-associated molecular patterns are expected to activate pattern recognition receptors such as Toll-like receptors (TLRs), resulting in the production of antiviral and pro-inflammatory cytokines [23]. When compared to healthy controls, patients with a CMV mononucleosis syndrome showed changed TLR2 and TLR7/8 responses and greater levels of pro-inflammatory cytokines interleukin 6 (IL-6) and tumor necrosis factor alpha (TNF-α), but not interleukin 10 (IL-10) [24]. These findings support previous evidence that CMV infection causes and/or amplifies inflammation with a synergistic mechanism between CMV infection and inflammation [25]. Furthermore, Hamilton et al. proposed an indirect route for CMV-induced fetal damage via placental elevation of pro-inflammatory cytokines and chemokines [26]. The association of fetal infection and its severity with an increase in pro-inflammatory cytokines involved in the Th1 immune response was recently confirmed by Bourgon et al. by investigating intra-amniotic cytokines in 40 CMV-infected fetuses following early maternal primary infection and 40 negative controls. In particular, a pattern of a specific increase in six proteins (IL-18soluble, TRAILsoluble, CRPsoluble, TRAILsurface, MIGinternal, and RANTESinternal) fitted severely symptomatic infection [27].

It has long been recognized that CD8+ T-cells play an important role in the regulation of CMV infection and illness [28,29]. CMV has evolved a multitude of anti-CD8+ T-cell responses, including the CMV downregulation of major histocompatibility (MHC) class I molecules occurring at all stages of viral replication [30,31]. On the other hand, the CD4+ T-cell response to CMV and its contribution to life-long carriage in healthy and immunocompromised patients is still to be fully understood [32,33]. It is uncertain how CMV genomes are maintained [34]. Similarly, we are just beginning to comprehend the viral and cellular processes that influence the development and maintenance of latent CMV infection, as well as reactivation from latency [35,36].

## 3. CMV Diagnosis

Clinical symptoms are more likely to be present in pregnant women who are infected for the first time than in women who have CMV reinfection or reactivation [2,22]. About 25% of pregnant women with primary CMV infection may have a flu-like syndrome with persistent fever as a prominent clinical feature, pharyngitis, fatigue, and myalgias [37]. Hepatosplenomegaly, cough, headache, rash, and gastrointestinal symptoms can rarely occur. Abnormal laboratory findings, mainly including lymphocytosis (30% to 50%) and slightly increased aminotransferase levels, are present in about 40% of the patients [37].

Primary CMV infection during pregnancy may be evidenced by serologic tests. CMV seroconversion, or the discovery of CMV IgG in a previously known nonimmune pregnant woman, is the gold standard for identifying primary infections [38]. CMV IgM and low-avidity IgG detection are useful as surrogate approaches for the serologic identification of primary infections [39]. Because CMV IgM antibodies may also be present during non-primary infections, they are not sufficient for diagnosing seroconversion.

CMV IgM can remain in the body up to 6 to 9 months after infection, and false-positive results can be caused by abnormal and nonspecific cross-reactive IgM (mostly from herpes simplex virus, and Epstein–Barr virus infections) or interference from rheumatoid factor or other autoimmune disorders. Studies comparing the sensitivities and specificities of immune–enzymatic assays for the detection of CMV IgM that utilized native versus recombinant antigens indicated that the latter exhibited lower sensitivities and specificities, most likely due to antigen misfolding by a prokaryotic production system [40]. The most significant drawbacks of the CMV IgG avidity test are due to variations in the ranges of low- and high-avidity thresholds across the commercial kits, and the extended persistence (>18 weeks) of low-avidity CMV IgG, which may result in a misdiagnosed primary CMV infection [39,41].

## 4. Prenatal Diagnosis

Currently, the diagnosis of invasive or noninvasive fetal infection may be provided to pregnant women with active CMV infection, particularly women with primary infection at high risk of CMV transmission. Amniocentesis and cordocentesis are invasive techniques, but ultrasound (US) and magnetic resonance imaging (MRI) are noninvasive [42]. The prenatal diagnosis of CMV infection can be easily performed through the detection of CMV DNA in amniotic fluid. Molecular diagnosis via PCR is currently favored over CMV culture due to its superior sensitivity (90 to 100%) [38].

Criteria for strengthening the negative predictive value of amniocentesis are as follows: 6 to 8 weeks after the predicted beginning of maternal infection, and 20 weeks of pregnancy, when CMV secreted from fetal kidneys into amniotic fluid is abundant, although it becomes detectable from the 16th week of gestation [12,43]. False-negative findings were reported, but it is to be considered that CMV can be transmitted in 10% to 15% of patients after amniocentesis, when the placenta becomes more permeable [22].

Cordocentesis allows for the assessment of fetal blood infections, CMV IgM antibodies, and other hematological and biochemical markers. Given that this invasive treatment involves a larger risk of adverse outcomes than amniocentesis, its value in the prenatal diagnosis of congenital CMV remains debatable [44].

In the absence of a serologic screening program for CMV infection in pregnancy, several investigations during the second or third trimester of pregnancy on US predictivity for clinical outcomes of infected fetuses found contradictory findings, indicating that the time of US examination and transitory fetal morphological traits may have a significant impact on the diagnosis [45,46].

When fetal intracranial abnormalities are found in US, MRI should be performed [47]. MRI has been proven to be more sensitive than US, and it could be useful to reveal cortical abnormalities even at 20–21 weeks, albeit the results may be more difficult to interpret and require professional neuroradiology advice [48].

## 5. Prenatal Prognostic Features

After a prenatal diagnosis of fetal CMV infection by amniocentesis, the prediction of serious sequelae is difficult. The date of infection may be helpful, while fetal abnormalities found via US and/or MRI are strongly correlated with the probability of symptomatic newborn estimation [45,46,47,48].

***Date of infection.*** Because most CMV infections, particularly non-primary infections, are asymptomatic, or symptoms may be undervalued, and because sequential serum testing for seroconversion is rarely performed, determining the timing of maternal infection can be challenging. Establishing the time of infection would be useful to assess the risk of congenital infection. This differs depending on the gestational age at which the initial infection develops [11]. In particular, infants infected during the first trimester had a larger proportion of cases with hearing and neurological sequelae than those later in pregnancy [9].

***Fetal anomalies.*** Early signs of a systemic illness can be seen in prenatal US as extra-cerebral abnormalities, since fetal brain involvement does not generally appear until many weeks later. There are three types of US findings: fetal cranial, extra-cranial, and placental/amniotic fluid abnormalities. Fetal cerebral abnormalities are the predominant US prognostic indication [2,22]. The negative predictive value of utilizing US for symptomatic infection at birth or termination of pregnancy at the time of prenatal diagnosis of congenital CMV infection is expected to be very low [45,46]. Such value can increase when combining US and MRI images.

***Laboratory markers.*** CMV DNA detection in blood, urine, saliva, and cervical-vaginal secretions may be positive in all or some samples. The absence of DNA, particularly in the blood, is related to the capacity of the immune system to retain the virus. Persistent maternal viremia predicts fetal infection and neonatal outcomes [49]. An early primary CMV infection may be followed by an abortion, while recurrent or persistent infection could be associated with recurrent abortion [50]. However, the lack of DNA in the blood of women with primary infection at their first test may not exclude the possibility that viral transmission to the fetus has occurred during the initial viremia [49].

The possible predictive role of the CMV viral load in the amniotic fluid is related to some variables such as gestational age at the time of amniocentesis and time after maternal infection. While the amniotic fluid CMV viral load in symptomatic fetuses may be higher than in asymptomatic fetuses, the overlap between these groups, as well as the effect of gestational age and timeframe between maternal infection and sample collection on the result, limit its clinical utility [51].

## 6. Prevention

A preventive CMV vaccination is desperately needed to prevent both congenital CMV illness and CMV sickness in immunocompromised people. There is preliminary evidence from Phase II studies that immunization can prevent CMV acquisition in seronegative women exposed to CMV in nature, but there are no Phase III results yet, although multiple candidate vaccines are being developed [52]. However, studies on animal models have raised concerns about the real effectiveness of these vaccines due to the different mechanisms of immune evasion of CMV [53,54].

Because CMV immunization is presently unavailable, antepartum screening, patient education, and the application of hygienic measures, which have been proven to be beneficial in the prevention of primary CMV infection, are the mainstays of prevention. Screening has not been adopted thus far because of a lack of understanding of congenital CMV infection as a serious health hazard among health and political authorities. Although CMV screening in pregnancy is common in Italy, very few pregnant women are told by their obstetricians about the sanitary precautions for avoiding transmission from children under the age of three. The German guidelines for the detection of viral infections in pregnancy recommend screening mothers at risk with family or professional CMV exposure during early pregnancy, as well as cryopreservation of the blood sample and storage for two years. Pregnant women should be tested for CMV antibodies at least twice (i.e., at 8–10 and 14–16 weeks of gestation) to detect seroconversion in previously seronegative women or reactivation/reinfection in previously seropositive women, as indicated by significant IgG titer increase and positive IgM antibodies [12]. When both IgG and IgM antibodies are positive in the initial test, IgG avidity can be used to distinguish recurring from primary infections. The double screening should be especially crucial in high-risk women, such as childcare workers who do not have children at home.

Appropriate hygienic precautions can prevent CMV transmission to seronegative pregnant women. The most aggressive preventive strategy requires counseling parents with seropositive children in daycare centers. Seronegative parents should be instructed in frequent handwashing after handling diapers and material contaminated with secretions, as well as warned about the potential risk of intimate contact, particularly mouth-to-mouth contact (i.e., kisses and sharing food, drinks, or flatware) [55]. These hygienic measures could also prevent CMV reinfection in seropositive women. Seronegative pregnant women should be advised to wear condoms if their partners are seropositive, as CMV can readily be acquired by reactivation in saliva or genital secretions [56].

## 7. Treatment

The absence of effective preventive measures such as a vaccine and the difficulty in applying preventive hygienic measures have led to the search for effective therapies that could reduce not only the rate of congenital infection but also the symptoms and sequelae of infected fetuses. Up to the present time, conservative care or pregnancy termination have been the treatment choices for congenital CMV infection. However, experimental interventions targeted at lowering the probability of maternal–fetal transmission or its severity have been proposed [57].

***Antiviral drugs***. The treatment of symptomatic newborns with ganciclovir or valganciclovir has presented rather favorable results. In 1994, a pilot study showed a better outcome in infants treated with intravenous ganciclovir for three months than in infants who were treated with ganciclovir for only two weeks [58]. Several studies or case reports confirmed that prolonged antiviral therapy is needed to maintain negative CMV replication for as long as possible. In fact, the treatment of symptomatic congenital CMV disease with valganciclovir for 6 months rather than 6 weeks did not enhance hearing in the near term but did appear to slightly improve hearing and developmental outcomes in the long run [59]. Antiviral therapy should likely be administered for longer than six months to infants with severe congenital CMV disease. In pregnancy, despite the favorable outcome reported in a few cases, the in vitro genotoxicity of ganciclovir did not allow for large and controlled studies [60,61,62].

A first controlled study using 8 g/24 h of oral valaciclovir showed a non-significant improvement in 21 CMV-infected fetuses treated at 27.4 weeks of gestation × 6.3 weeks, because a non-significant good outcome occurred in 48% of patients vs. 42% in controls [63]. However, a subsequent uncontrolled open-label trial reported the efficacy of 8 gr. daily valaciclovir for pregnant women bearing a mildly symptomatic CMV-infected fetus: 82% of 21 symptomatic fetuses treated at 26.9 weeks × 89 days had a good outcome compared to 43% from the literature metanalysis [64]. This therapy was safe and associated with a significant decrease in the viral load and an increase in the platelet count in fetal blood. A favorable efficacy of the 8 g/day valaciclovir was confirmed through a randomized, double-blind, placebo-controlled trial in 90 pregnant women with CMV infection acquired early in pregnancy [65]. In 45 patients treated with valaciclovir, 11% of amniocenteses were CMV positive, compared to 30% in the placebo group (*p* = 0.027), resulting in an odds ratio of 0.29 (95% CI 0.09–0.90) for decreased vertical transmission. After treatment of 12 pregnant women with 8 g/day valaciclovir until amniocentesis or until delivery, if the fetus was infected, De Santis et al. reported a transmission rate of 17% at amniocentesis, and 42% at birth. At a follow-up of 5–28 months, one infant developed sensory-neural hearing loss [66]. A further study supported the preventive role of high-dosage valaciclovir, comparing 65 treated pregnant women and 65 selected controls who initiated treatment at a median gestational age of 12.71 weeks for a median duration of 35 weeks. On multivariate logistic regression, fetal infection was lower in the treated group (odds ratio, 0.318 (95% CI, 0.120–0.841); *p* = 0.021) [67]. However, in vitro studies by Hamilton et al. reported viral inhibition in CMV-infected placental explants with letermovir (83.3%), maribavir (83.6%), cidofovir (89.3%), and ganciclovir (82.4%), but not with acyclovir [68].

One of the authors (GN), who is frequently requested for counseling by pregnant women with primary CMV infection (www.cmvcongenital-nigro.com; www.giovanninigro.it; www.anticito.org), followed 13 women who were treated with valaciclovir; 10 soon after the diagnosis of primary CMV infection for the prevention of maternal–fetal CMV transmission, and 3 after a CMV-positive amniocentesis.

As shown in Table 1, in the prevention group,
-Two patients were treated for the prevention of fetal infection and had negative amniocentesis and neonates, while another one delivered an infected infant with slightly abnormal MRI images but a favorable outcome associated with valganciclovir therapy.-Six patients had CMV-positive amniocentesis: one opted for termination of pregnancy (TOP), another one had an asymptomatic neonate, two had two infants with right deafness, and another one delivered an infant with bilateral deafness and cortical malformations.-One woman refused amniocentesis and was also treated with HIG. The infant was not infected.

Of the three patients who were treated after CMV-positive amniocentesis,
-One (also treated with HIG) had an infant with a favorable outcome.-One had psychomotor delay.-One had an infant with right deafness and neurological complications.

This case series, although not comparable to a large-scale study because of the low number of patients and non-randomized enrolment, suggests that valaciclovir, even at a high and prolonged dosage, may not prevent CMV infection and disease.

***CMV hyperimmune globulin.*** Since 1952, immunoglobulin therapy via regular infusions is a lifesaver in patients who are born with agammaglobulinemia and other congenital or postnatally acquired immune deficiencies. Immunoglobulins were subsequently labeled or “off label” used for many other diseases, including infections, with uncommon minor or major side effects [69].

In 1999, the possible protective role of anti-CMV specific hyperimmune globulin (HIG) from the vertical transmission of CMV and severe CMV-related sequelae was reported in a pregnant woman with intra-uterine growth restriction of one twin fetus, following a primary CMV infection in the first trimester [70]. After a few case reports confirming the protective role of HIG, a prospective non-placebo randomized but controlled study investigated the possible efficacy of HIG for the treatment or prevention of congenital CMV disease in 68 pregnant women with confirmed primary CMV infection [71]. Controls were women who refused HIG after a negative ultrasound evaluation. Monthly intravenous administration of CMV HIG at the dosage of 100 Units/kg of maternal weight reduced the rate of CMV infections from 40% to 16% compared to the control group (*p* = 0.02). Moreover, 200 Units/kg of HIG for therapy of infected or injured fetuses was associated with a significantly lower risk of congenital CMV disease (adjusted odds ratio, 0.02; 95 percent confidence interval, −∞ to 0.15; *p* < 0.001) compared to untreated controls [71]. Numerous subsequent studies supported the efficacy of HIG for preventing congenital CMV infection or disease [72,73,74,75,76,77,78,79,80,81,82].

In 2014, a randomized trial by Revello et al. compared monthly infusions of 100 Units/kg HIG versus saline solution [79]. Results showed a trend in favor of HIG treatment, but the difference between the two groups was not statistically significant (30 vs. 44%; *p* = 0.13). Hughes et al., in an uncompleted randomized trial including many subjects enrolled after only detection of low anti-CMV IgG avidity, via monthly administration of 100 Units/kg of HIG before 24 weeks gestation, found that the patients, compared to controls, showed a significant occurrence of side effects and an abnormally low rate of CMV transmission (22.7% in HIG-treated vs. 19.4% in untreated patients) [83]. However, while Revello’s trial showed a possible efficacy of HIG in decreasing CMV transmission in pregnancy with non-significant occurrence of obstetrical events, Hughes et al. first showed that, contrary to hundreds of papers concerning non-specific immunoglobulin and hyperimmune globulin, it is more dangerous to human beings than a viral infection [79,83]. Both trials by Revello et al. and Hughes et al. used low-dosage HIG at an interval too long between infusions, and received critical observations [84,85].

Recent studies strongly support the efficacy of HIG in preventing CMV transmission and disease in pregnancy. A non-randomized but controlled study concerning 304 pregnant women with primary infection, who delivered 108 CMV-infected infants, showed that high-dose HIG (200 U/kg) is associated with a decreased prevalence and copy/number of maternal DNAemia and may prevent fetal infection and disease [85]. Of the 90 HIG-treated women in the first trimester (56 had seroconversion), 1 aborted after fetal intestinal hyperechogenicity and another 1 had an infant with bilateral deafness (2.2%), compared to 25 of 91 (27.5%) non-treated mothers (46 had seroconversion), who had 5 symptomatic TOPs and 20 symptomatic infants at a long-term follow-up (*p* < 0.001) [49]. Very favorable results were obtained by Kagan et al., who first reported a transmission rate of 7.5% in 40 pregnant women with primary CMV infection in the first trimester after biweekly 200 IU/kg HIG up to 20 weeks gestation, and then a maternal–fetal transmission in 6.5% in the 153 fetuses of 149 HIG-treated women [86,87].

Table 2 reports our unpublished and not-peer-reviewed data about the efficacy of biweekly administration of high-dosage HIG: CMV transmission occurred in 4/32 patients (12.5%), 1 of whom opted for TOP. The two-year outcome of all three infected infants was unaffected. Notably, (1) all 4 patients who transmitted the virus had a seroconversion; (2) 2/3 patients with fetal transmission had positive CMV DNAemia; and (3) none of the 17 patients (53%) with presumed immediately post-conceptional infection did not transmit CMV to the fetus. Comparing these results with those in Table 1, which were obtained in a small group of selected patients, the infants born to women treated with HIG showed significantly lower rates of CMV DNA positivity in urine (9.7% vs. 75.0%; *p* < 0.0001) and abnormalities after follow-up (0.0% vs. 41.7%; *p* < 0.0001).

Except for the trials by Revello et al. and Hughes et al., the safety of HIG infusions was shown by all other reports [69]. In particular, a study including 358 women with a primary CMV infection during pregnancy, 164 of whom received one or more infusions of HIG, showed that receiving multiple doses of HIG (range 1 to 8) was significantly correlated with an increase in birth weight (*p* = 0.006) and gestational age at delivery (*p* = 0.014) [88].

Passive immunization remains an important treatment modality to provide the immediate benefit of protective antibody levels. For passive immunization with polyclonal therapies, the antibody source can be human or animal plasma. One of the most well-established and proven platforms for passive immunization is HIG, with more than 20 FDA-approved products to address a broad range of targets or pathogens [69]. Contrary to the antiviral drugs only capable of inhibiting CMV replication, HIG has also immunomodulatory effects due to the content of IgG antibodies blocking the fetal cell receptors for CD8+ T-cells, NK cells, and cytokines, some of which are toxic to the developing brain [12,89].

## 8. Conclusions

Congenital CMV infection is still one of the predominant causes of severe and permanent disabilities in children: it can lead to severe neurological sequelae, especially if acquired in embryonic life. Despite some attempts, a reliable vaccine against CMV is still not available. Therefore, preventive measures, raising awareness among health workers and pregnant women, and good health education remain the preventive pillars of primary CMV infection in pregnancy. The importance of congenital CMV infection in public health is not well known. Routine CMV serologic screening in pregnant women is not yet established, although it is frequently performed in a few countries [90]. On the other hand, specific treatments for primary CMV infection in pregnancy or fetal infection have not been officially recognized. Although not confirmed by two randomized trials, many randomized animal studies, controlled human trials, and case reports have shown that HIG could be safe and capable of preventing maternal–fetal CMV transmission and, mostly, decreasing fetal or neonatal disease. HIG is the natural approach to the prevention of congenital CMV disease, since it contains high-titer and avidity anti-CMV antibodies and immunomodulatory activities, decreasing the intensity of the inflammatory response to CMV infection and subsequent tissue damage. While waiting for further studies, there is enough evidence to support the use of passive immunization for the prevention of congenital CMV disease. Oral valaciclovir could also have a promising role in the prevention of congenital CMV infection, but information currently available on the efficacy and safety of high-dosage and prolonged antiviral treatment is based on few studies. The effectiveness of valaciclovir might be evaluated by a randomized trial comparing this drug versus HIG. Given the different mechanisms of activity, the combined administration of an antiviral and HIG could also be evaluated.

## Figures and Tables

**Table 1 viruses-15-01376-t001:** The outcome of pregnant women with primary CMV infection before 14 weeks’ gestation who were treated with valaciclovir to prevent fetal infection or disease.

Patient Number Years	wg Maternal Infection	wg CMV DNA Genomes/mL in Blood	wg Start Valaciclovir (wg from Maternal Infection)	wg CMV DNA Genomes/mL in Amniotic Fluid	wg Fetal Abnormalities by Ultrasound and/or Magnetic Resonance Images (MRI)	wg Delivery	Neonatal CMV DNA Genomes/mL in Urine Sex Birth Weight Clinical and MRI Features	Outcome (Months of Follow-Up)
**1–32**	5	10: negative	11 (6)	21: 1,662,560	no	CS 39	>90,000,000 F 3460 Right deafness	VGC, Right deafness (60)
**2–28**	5	12: 270	12 (7)	21: 500,330	no	CS 38	>50,000,000 M 3380	Unaffected (2)
**3–29**	14	18: <1000	18 (4)	21: 3271	no	CS 38	60,000 M 2920 Leukodystrophy	VGC, Unaffected (12)
**4–31**	7	13: negative	13 (6)	19: 716,430	26: Microcephaly Leukodystrophy Polymicrogyria	VD 38	5,389,571 M 2640 Microcephaly, Ventriculomegaly, Polymicrogyria, Pachygyria	VGC, Bilateral deafness, Psychomotor delay (8)
**5–33**	6	12: negative	21 (15)	20: 2,460,000	no	CS 37	38,500,000 M 2930 Ventriculomegaly, Temporal cysts, Right deafness	VGC Right deafness, West syndrome, Psychomotor delay (36)
**6–29**	5	13: negative	14 (9)	21: 4,608,272	no	TOP	NA	NA
**7–32**	4	8: 319	8 (4)	20: negative	no	VD 35	3,384,488 F 2910 Leukodystrophy, Periventricular cysts	VGC, Unaffected (8)
**8–35**	11	19: negative	21 (10)	21: 758,515	no	VD 39	23,318,512 F 2600	Unaffected (3)
**9–30**	10	16: 118	16 (6)	22: 2,060,000	no	CS 39	>10,000,000 M 2855 Ventriculomegaly, Right deafness	VGC, Right deafness (5)
**10–28**	8	NP	11 (3)	NP	no	VD 38	Negative M 3090	Unaffected (1)
**11–34**	6	10: 480	12 (6)	20 negative	no	VD 39	Negative F 3220	Unaffected (4)
**12–26**	3	7: negative	22 (19)	21: 1278	no	VD 37	14,000,000 M 3560 Leukodystrophy, Ventriculomegaly	Psychomotor delay (8)
**13–28**	7	10: 640	12 (5)	20: negative	no	VD 40	Negative F 2815	Unaffected (8)

F: female; FU: follow-up; M: male; TOP: termination of pregnancy; VGC: valganciclovir; WG: weeks of gestation.

**Table 2 viruses-15-01376-t002:** The outcome of pregnant women with primary CMV infection before 14 weeks’ gestation who were treated with 200 U/kg hyperimmune globulin (HIG) biweekly to prevent fetal infection or disease *.

Patient Number Years	wg Maternal Infection	wg CMV DNA Genomes/mL in Blood	wg Start HIG (wg from Maternal Infection)	wg CMV DNA Genomes/mL in Amniotic Fluid	wg Fetal Abnormalities by Ultrasound and/or Magnetic Resonance Images (MRI)	wg Delivery	Neonatal CMV DNA Genomes/mL in Urine Sex Birth Weight Clinical and MRI Features	Outcome (Years of Follow-Up)
1–34	4	11 negative	12 (8) 14-16	NP MRI negative	no	VD 41	Negative M 3500	Unaffected (8)
2–30	3	8: 776	11 (8) 13-15	19 negative	no	VD 38	Negative F 3400	Unaffected (9)
3–38	9	14: 111	15 (6) 17	20 negative	no	VD 38	Negative F 2785	Unaffected (6)
4–35	3	7: 960	9 (6) 11-14-16	19 negative	no	VD 39	Negative F 3700	Unaffected (6)
5–39	3	9: 121	11 (8) 13-16	18 negative	no	VD 39	Negative F 3360	Unaffected (4)
6–33	4	7: 308	9 (5) 11-14-16	19 negative	no	VD 42	Negative F 3988	Unaffected (3)
7–33	11	15: 30	15 (4) 17	20 negative	no	VD 39	Negative M 3020	Unaffected (2)
8–37	3	7 negative	9 (6) 11-13-15-17-19	20 negative	no	VD 38	Negative M 3,3700	Unaffected (4)
9–31	4	11 negative	11 (7) 13-15-17-19	20 negative	no	VD 38	Negative F 3165	Unaffected (3)
10–38	3	10 negative	11 (8) 13-15	NP MRI negative	no	VD 40	Negative F 3450	Unaffected (5)
11–37	3	8 negative	11 (8) 13-16	20 negative	no	VD 41	Negative M 3800	Unaffected (7)
12–38	4	10: 818	13 (9) 15-18-20	21 negative	no	VD 41	Negative M 3500	Unaffected (2)
13–35	5	10 negative	14 (9) 16-18	NP MRI negative	no	VD 41	Negative M 3200	Unaffected (7)
14–37	9	13: 60	15 (5) 17-19	NP MRI negative	no	VD 40	10,000,000 F 2910 Unaffected	Unaffected (4)
15–33	9	14 negative	14 (5) 16-18	21 negative	no	CS 38	Negative F 2600	Unaffected (7)
16–39	6	9: 690	10 (4) 13-15-17	19 negative	no	VD 40	Negative M 3500	Unaffected (9)
17–34	4	8 negative	8 (4) 11-13-15-17	20 negative	no	VD 39	Negative F 3220	Unaffected (4)
18–38	7	11 negative	12 (5) 14-16-18	NP MRI negative	no	VD 38	Negative F 2960	Unaffected (7)
19–35	13	17: 8178	16 (3) 18-20	NP MRI negative	no	VD 38	Negative F 2690	Unaffected (5)
20–32	4	11: 1837	12 (8) 14-17	NP MRI negative	no	VD 37	Negative M 3600	Unaffected (4)
21–27	7	10: 906	13 (4) 15-17	20 negative	no	CS 38	Negative F 3160	Unaffected (8)
22–30	6	NP	12 (6) 14-17	20 negative	no	VD 39	Negative F 3110	Unaffected (7)
23–38	3	6: 61,000	10 (7) 13-16-18	20 negative	no	VD 37	Negative F 2740	Unaffected (3)
24–35	9	14: <1000	15 (6) 18-20	20: 39,420	no	CS 39	11,842,000 M 3090 Unaffected	Unaffected (9)
25–25	3	8 negative	10 (3) 12-14-16	18 negative	no	CS 38	Negative M 3200	Autism CMV negative (8)
26–39	4	10 negative	12 (8) 14-16-18	20 negative	no	VD 39	Negative M 3450	Unaffected (3)
27–33	4	10 negative	11 (7) 13-16	18 negative	no	VD 38	Negative M 3920	Unaffected (5)
28–33	5	9: 1461	12 (7) 15-17-19	20 negative	no	VD 37	Negative F 3150	Unaffected (7)
29–30	4	9 negative	9 (5) 12-15-18	NP MRI negatives3	no	CS 41	Negative 2640	Unaffected (4)
30–34	7	12: 154	12 (5) 15-18-21	NP MRI negative	no	VD 38	Negative 2970	Unaffected (8)
31–36	7	NP	14 (7) 16-18	19: 1,340,000	18 wg: ascites, intestinal hyperecogenicity	TOP	NA	NA
32–42	9	14 negative	15 (6) 17-19	NP MRI negative	no	VD 41	17,587,502 M 3480 Unaffected	Unaffected (7)

F: female; M: male; TOP: termination of pregnancy; VGC: valganciclovir; WG: weeks of gestation. * The information presented in this table has not been peer reviewed.

## Data Availability

The data presented in this study are available from the corresponding author upon request.

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
