# Peer review of "Prevention of Congenital Cytomegalovirus Infection: Review and Case Series of Valaciclovir versus Hyperimmune Globulin Therapy"

_viruses, 2023, doi:10.3390/v15061376_

Round 1

Reviewer 1 Report

The paper by Nigro el al is an excellent comprehensive review of nearly all aspects of the causes, diagnosis and treatment of congenital CMV infection and will appeal to readers who seek a detailed literature  review and introduction to this important topic.

Tables 1 and 2 which are  difficult to follow due to the large number of abbreviations used.  These tables should be simplified or better yet condensed into a single summary table.

There is also extensive overlap between tables 1 and 2 ant the descriptive text.

English grammar and syntax need improvement in many places.

English grammar and syntax need improvement in many places.

Author Response

REFEREE 1

We are very grateful to the reviewer for the time dedicated and appreciation of our work. Following his suggestions, we changed the paper as indicated below:

  • Almost all abbreviations were eliminated in both Tables.
  • We acknowledge that there is an overlapping between Tables and descriptive text. However, we just summarized the main data in the text, while the Tables include detailed and important findings (i.e. prenatal and neonatal manifestations, valganciclovir therapy) concerning the patients in relation to their outcomes, which are not reported in the text.
  • We tried to improve the English language as much we could. However, we would appreciate very much if the reviewer was so kind to indicate briefly the places needing improvement of English grammar and style

Reviewer 2 Report

The paper "Prevention of Congenital Cytomegalovirus Infection: Review 2 and Case-Series of Valaciclovir Versus Hyperimmune Globulin 3 Therapy" was quite enlightening. CMV is a very important cause of primary and recurrent maternal infection, and its importance only increases in discussions on congenital infections. Thus, the manuscript was of great interest and importance.  The manuscript is well written, and the results were very convincing in regards to the effects of HIB on maternal CMV infections. Experimental approach was appropriate, and the conclusion is supported by the results. Overall, the manuscript is appropriate for publication.

However, certain minor issues need to be addressed prior to publication. For example, "However, the 53 majority of infected neonates acquire CMV from seropositive mothers, who are or seronegative..." (line 53-55) should be adjusted for clarity. Additionally, the use of the word "human" in "the relationship with CMV is based on human’s intrinsic, innate, and adaptive immune responses..." (line 121-122) should be altered to have better language. Additional discussion on the ramifications of HIB as it compares to prior studies, and its impact, should be provided.

as above

Author Response

REFEREE 2

We are very grateful to the reviewer for the time dedicated and appreciation of our work. Following his suggestions, we changed the paper as indicated below:

  • Page 3. Lines 5-8: However, the majority of infected neonates acquire CMV from seropositive mothers, who are considered to be immune, or seronegative women who had a primary infection in the second half of pregnancy, and have a low prevalence of symptoms, both at birth and later on [5,6].
  • Page 5. Lines 9-12: This sentence has been changed in this way: The relationship with CMV is based on intrinsic, innate, and adaptive immune responses of human beings, as well as numerous viral countermeasures, which collectively enable a dynamic balance between host and pathogen that largely prevents disease while not completely eliminating the virus from the human body.
  • Page 14. Lines 6-13: Additional discussion concerning effectiveness and impact on primary CMV infection in pregnancy was included as follows: Although not confirmed by two randomized trials, many randomized animal studies, controlled human trials and case reports showed that HIG could be safe and capable of preventing maternal-fetal CMV transmission and, mostly, decreasing fetal or neonatal disease. HIG is the natural approach to the prevention of congenital CMV disease, since it contains high titer and avidity anti-CMV antibodies and immunomodulatory activities decreasing the intensity of the inflammatory response to CMV infection and subsequent tissue damage. Waiting for further studies, there is enough evidence to support the use of passive immunization for prevention of congenital CMV disease.

And the last sentence:

The effectiveness of valaciclovir might be evaluated by a randomized trial comparing this drug versus HIG. Given the different mechanism of activity, the combined administration of an antiviral and HIG could also be evaluated.